# Evaluation of a 'serious game' on nursing student knowledge and uptake of influenza vaccination

**Gary Mitchell** *, **Laurence Leonard, Gillian Carter, Olinda Santin, Christine Brown Wilson**

School of Nursing and Midwifery, Queen's University Belfast, Belfast, United Kingdom

\* Gary.Mitchell@qub.ac.uk

## Abstract

**Data Availability Statement:** All relevant data are within the manuscript and its Supporting Information files.

**Funding:** The author(s) received no specific funding for this work.

### Background

Influenza is a serious global healthcare issue that is associated with between 290,000 to 650,000 deaths annually. The aim of this study is to evaluate the effect of a 'serious game' about influenza, on nursing student attitude, knowledge and uptake of the influenza vaccination.

### Methods

1306 undergraduate nursing students were invited, via email, to play an online game about influenza between September 2018 and March 2019. 430 nursing students accessed the game and completed an 8-item questionnaire measuring their attitudes to influenza between September 2018 and March 2019. In April 2019, 356 nursing students from this sample completed a follow-up 2-item questionnaire about their uptake of the influenza vaccination. A larger separate 40-item knowledge questionnaire was completed by a year one cohort of 124 nursing students in August 2018 prior to receiving access to the game and then after access to the game had ended, in April 2019. This sample was selected to determine the extent to which the game improved knowledge about influenza amongst a homogenous group.

### Results

In the year preceding this study, 36.7% of the sample received an influenza vaccination. This increased to 47.8% after accessing to the game. Nursing students reported perceived improvements in their knowledge, intention to get the vaccination and intention to recommend the vaccination to their patients after playing the game. Nursing students who completed the 40-item pre- and post-knowledge questionnaire scored an average of 68.6% before receiving access to the game and 85.2% after. Using Paired T-Tests statistical analysis, it was determined that this 16.6% increase was highly statistically significant (P < 0.001).

**Competing interests:** The authors have declared that no competing interests exist.

## Conclusions

The research highlights that the influenza game can improve knowledge and intention to become vaccinated. This study suggests that improvement in influenza knowledge is likely to encourage more nursing students to receive the influenza vaccination.

## Introduction

Influenza is a serious global healthcare issue and there are approximately one billion cases annually [1]. Influenza is also associated with between 290,000 to 650,000 deaths every year [1–3]. Seasonal influenza is a substantial cause of a large number of lower respiratory tract infections like pneumonia or bronchitis [4]. The World Health Organisation (WHO) have prioritised influenza as a key area for future development, in their recently published global strategy [1]. In this document, the WHO have outlined a number of recommended actions which relate to supporting influenza prevention, control and preparedness internationally. A key influenza prevention strategy is promoting the annual uptake of the influenza vaccination amongst at-risk populations and healthcare professionals [5,6].

Encouraging influenza vaccination uptake amongst healthcare professional groups has been a key prevention strategy for many years [5–7]. The annual vaccination is recommended by global healthcare institutions due to its role in reducing influenza and transmission between patients [8–11]. The vaccination also reduces the occurrence of healthcare professional absenteeism, subsequent staff shortages and reduced quality of care [12–14]. Despite this, influenza immunisation uptake can be challenging with vaccinations being administered to approximately 74% of front-line healthcare professionals in England [15] and 80% of front-line workers in the USA [16].

There have been a number of research studies which have examined healthcare professional views on influenza vaccination and two important themes have emerged as to why healthcare professionals do not receive the influenza vaccination [17]. The first reason relates to healthcare professional misconception about the influenza vaccination, its effectiveness and safety [17–26]. The second reason is around healthcare professional ability to access the free vaccination [17–21]. In response to this, there have been a range of interventions utilised to increase vaccination uptake amongst healthcare professionals including the use of educational strategies, organisational flu campaigns, incentivisation and adoption of vaccination champions [22–30].

Nursing students are one group that are at a high risk of exposure to seasonal influenza. Despite this, they are a marginalised group and are often absent in practice guidelines about influenza and empirical research investigation on the topic [31]. In the UK, current National Health Service (NHS) advice about who should receive the influenza vaccination, does not explicitly mention nursing students [32]. To our knowledge, there are no national or international recommendations for nursing students about the influenza vaccination. As a consequence, knowledge and subsequent uptake of influenza vaccination is potentially low amongst nursing students. Research indicates that between15% to 50% of nursing students will receive annual influenza vaccination, however due to the paucity of research in this area, combined with small numbers of participants in these published studies, the data is not generalisable [31–35]. While research is limited, it is postulated that the rationale for low influenza vaccine uptake amongst healthcare professional students is due to limitations in knowledge, professional misconceptions and ease of access [31–37].

An increasingly effective and innovative method of educating professionals is through the use of 'serious games' [38–40]. Unlike traditional entertaining games, the 'serious game' has been designed with a specific educational purpose in mind. The 'serious game' is considered as an entertaining tool with a purpose of education, where players cultivate their knowledge and practice their skills through gaming [38–45]. In the context of infection control and prevention, there has been evidence to suggest that a 'serious game' can enhance healthcare education and practice [46–48].

The aim of this study was to evaluate the effect of a 'serious game' about influenza, on nursing student attitude, knowledge and uptake of the influenza vaccination. The objectives of this study were as follows:

1. To examine nursing student perceptions about their attitudes and understanding of influenza before and after playing the game.

2. To learn if playing the 'serious game' increased nursing student knowledge about influenza.

3. To establish if playing the 'serious game' correlated with increases in flu vaccination uptake amongst nursing students.

## Methods

### Ethics

This study was approved by the School of Nursing and Midwifery's Research Ethics Committee at Queen's University Belfast in July 2018 (2.GMitchell 09.18M1.V1). Participants did not provide verbal or written consent but were informed that they were under no obligation to complete any of the questionnaires. Participants gave their consent to complete the questionnaire when they actively accessed the survey web links.

### Design, setting and population

This study took place at one university in Northern Ireland. All university students (n = 1306) who were undertaking a BSc Honours Degree in Nursing were provided with access to the 'serious game' between 1st September 2018 and 31st March 2019. The nursing students were enrolled in one of the four programmes; adult nursing, mental health nursing, children's nursing or learning disability nursing. All students received access to the same version of the 'serious game'.

In total 430 nursing students, from year 1, 2 and 3 (32.92%), played the game completed a short 8-item questionnaire about their perceptions and attitudes to influenza throughout September 2018 to March 2019. From this sample of 430 nursing students, 356 (82.79%) went on to complete a further 2-item questionnaire about their uptake of the influenza vaccination, after access to the game had ended, in April 2019.

In addition to these two short questionnaires, we conducted a separate 40-item pre and post knowledge questionnaire with a homogenous sample of first year adult nursing students (n = 145) to determine if knowledge about influenza improved in this cohort. Year one students were selected on the basis that they were likely to have less knowledge about influenza compared to their peers in year two and year three. The year one students in this sample had received twelve weeks of standard nurse education at the university and undertaken two clinical placements, each lasting 6 weeks, as part of their nursing programme prior to completing a pre-knowledge questionnaire about influenza in August 2018. A follow-up post-knowledge questionnaire about influenza was administered to this cohort in April 2019. In total 124 students (85.52%) completed both knowledge questionnaires.

## Intervention

The game, known as 'Flu Bee Game' was developed by Focus Games Ltd in 2017 [49]. The game has been used to promote knowledge about influenza and encourage vaccination uptake amongst healthcare professionals at multiple international settings [49,50]. The game has been associated with improving vaccination uptake in multiple settings but has yet to be tested amongst nursing students [50].

The Flu Bee Game is an HTML5 web application with a supporting website. The game works on any device through a web browser and only takes a few minutes to play. Players answer random questions, from an existing question bank, about influenza and vaccination. If they get a question correct, they build a 'honeycomb path' that leads them to 'Queen Bee' status. Players of the Flu Bee Game can share their success on the game's leader board and invite colleagues to play via social media. The serious game presents players with influenza facts and challenges common myths associated with vaccine hesitancy. These common myths include statements like "I'm healthy and so do not need a flu vaccine", "I'm 12 weeks pregnant so cannot have the flu vaccine" or "I don't want to risk getting the flu from the vaccine" [49,50].

The Flu Bee Game takes approximately 90 seconds to play and players can have multiple attempts, as questions are randomly generated. Players receive feedback and further information on each question they answer in the game. The overall objective of this serious game is to create awareness about influenza, dispel myths associated with the influenza vaccine and increase uptake the vaccination.

## Consent and recruitment

All students (n = 1306) received information about the 'serious game' and this study by a person unrelated to the project in August 2018 via email. Students were provided with a web-link to the game and informed that they could access the game any time throughout September 2018 to March 2019. All students from years 1 to 3 could also opt to complete a voluntary 8-item questionnaire immediately after playing the game. All students received three follow-up emails, at the end of September 2018, the end of November 2018 and the end of January 2019, to remind them of the availability of the 'serious game' and accompanying questionnaire. These reminders were also sent by a person unrelated to the project. Students did not provide any personal or demographic information in their responses but could report their university email address to be contacted about a further 2-item questionnaire, about their uptake of influenza vaccination, in April 2019. A second 2-item questionnaire was emailed to all nursing students who had played the game, completed the first questionnaire and provided their email address.

In addition to these questionnaires, the authors worked with Focus-Games Ltd to design a 40-item knowledge questionnaire about the myths associated with influenza and influenza vaccination. Face validity was tested with 12 nursing students, who were not part of the cohort, prior to administration. A homogenous group of year one nursing students were invited to complete this 40-item questionnaire before they received access to the 'serious game' in August 2018 and then again in April 2019 once access to the game had ended. This cohort of year one nursing students received both questionnaires via email by a person unrelated to the study.

Students did not have to sign written consent forms but were informed that they were under no obligation to complete any of the questionnaires. It was assumed that students gave their consent to complete a questionnaire when they actively accessed the survey web links. Student participants were required to use their own laptop, computer tablet or mobile phone to complete the questionnaires. Questionnaires were completed by students in their own time and not during any timetabled classes.

## Data collection

All nursing students (n = 1306) were eligible to participate in the first 8-item questionnaire. This questionnaire, designed by the authors, sought to examine nursing student perception of their knowledge about influenza, likelihood of getting vaccinated and importance of promoting the vaccine amongst their patients after playing the game. This was achieved using Likert scale items with participants asked to select an option for each question; ranging from very poor, poor, average, good to very good. In total 430 nursing students (32.92%) completed this questionnaire, then in April 2019, 401 nursing students from this sample agreed to be contacted via email to receive a follow-up 2-item questionnaire about their uptake of influenza vaccine. Subsequently 356 nursing students went on to complete this second questionnaire.

Finally, a cohort of year one nursing students (n = 145) were purposely selected to participate in a pre- and post-knowledge questionnaire about influenza after receiving access to the 'serious game'. Overall, 124 nursing students (85.52%) completed a 40-item pre and post questionnaire, designed by the authors, about myths associated with influenza and influenza vaccination. All nursing students that completed both the pre and post knowledge questionnaires were automatically entered into a prize raffle and three winners received a complimentary stay at a hotel in Belfast.

While other validated measures were available, the authors developed their own questionnaires to answer the research question. The rationale for this was due to a combination of factors including the nature of the intervention (a digital game), its duration (each play taking approximately two minutes), the sample (nursing students) and the information the authors wanted to glean.

## Data analysis

Descriptive statistics were used to illustrate the findings from the 8-item questionnaire and the 2-item questionnaire to measure nursing student perceptions about their attitudes and understanding of influenza and subsequent vaccination uptake. The pre and post knowledge questionnaires, administered to the cohort of nursing students, were analysed using paired t-tests to establish if the 'serious game' increased student knowledge about influenza.

## Results

### 8-Item questionnaire about attitudes and perception

The 8-item questionnaire measuring influenza attitudes and perceptions, was completed by 430 nursing students. Of the respondents who completed the 8-item questionnaire 53.3% were from year one (n = 229), 24.4% were from year two (n = 105) and 22.3% were from year three (n = 96). Of these participants, 36.7% (n = 158) had received the influenza vaccine the year prior and 63.3% (n = 272) had not received the vaccination to date.

Nursing students perceived that their knowledge about influenza and the vaccination was very good (n = 36/8.37%), good (n = 164/38.14%), average (n = 188/43.72%), poor (n = 32/7.44%) or very poor (n = 10/4.30%) prior to playing the game. Immediately after completing the game, nursing student perception of their knowledge increased with 91.4% of students perceiving their knowledge to be either good (n = 187/43.49%) or very good (n = 206/47.91%).

In relation to their willingness to receive the influenza vaccination, 39.7% (n = 171) stated that prior to playing the game they did not intend to receive the vaccination. After playing the game, this number decreased to 7.4% (n = 32). The number of students who stated they definitely would receive the influenza vaccination doubled from 29.5% (n = 127) pre-game, to 58.6% (n = 252) post-game.

**Table 1. 8-Item questionnaire about attitudes and perception.**

| Questionnaire Item | Very Poor | Poor | Average | Good | Very Good | | Total |
|---|---|---|---|---|---|---|---|
| Before Playing the Flu Game, How Did You Rate Your Knowledge? | 10 | 32 | 188 | 164 | 36 | | **430** |
| After Playing the Flu Game, How Did You Rate Your Knowledge? | 2 | 9 | 26 | 187 | 206 | | **430** |
| | **Definitely Would Not** | **Probably Would Not** | **Probably Would** | **Definitely Would** | | | |
| Before Playing the Flu Game, How Likely Were You to Get Vaccinated? | 30 | 141 | 132 | 127 | | | **430** |
| After Playing the Flu Game, How Likely Are You to Get Vaccinated? | 7 | 25 | 146 | 252 | | | **430** |
| | **Very Important** | **Moderately Important** | **Quite Important** | **Slightly Important** | **Not Important** | | |
| Before Playing the Flu Game, How Important Did You Think It Was To Encourage Your Patients and The Public To Receive The Vaccination? | 189 | 107 | 93 | 32 | 9 | | **430** |
| After Playing the Flu Game, How Important Did You Think It Was To Encourage Your Patients and The Public To Receive The Vaccination? | 358 | 58 | 12 | 1 | 1 | | **430** |
| | **Feb-16** | **Sep-16** | **Feb-17** | **Sep-17** | **Feb-18** | **Sep-18** | |
| What Nursing Cohort Do You Belong To? | 19 | 77 | 27 | 78 | 77 | 152 | **430** |
| | **Yes** | **No** | | | | | |
| Did You Receive the Flu Vaccine Last Year? | 158 | 272 | | | | | **430** |

Finally, as it pertains to student perception about the importance of promoting the influenza vaccination to their patients and public, 44.0% (n = 189) felt this was very important pregame and this increased to 83.3% (n = 358) post-game. Less than 1% of the sample (n = 2) believed the influenza vaccination was either slightly important or not important post-game.

Descriptive statistics from this 8-item questionnaire can be viewed in Table 1.

## 2-Item questionnaire about influenza vaccine uptake

In total 356/401 nursing students (88.8%) went on to complete the second questionnaire. The first questionnaire item determined if the respondent had received the influenza vaccination during the period of 1st September 2018 to 31st March 2019. Overall, 47.8% (n = 170) of students who played the game received the vaccination and 52.2% did not (n = 186). Of the students who did not receive a vaccination they were asked to complete the second item of the questionnaire to provide a reason for their decision. The most common reason selected by nursing students was related to a lack of time (33.9%). The remaining reasons were confusion about where to receive vaccination (23.7%), concerns about receiving the vaccination (21.5%), forgetting to get vaccination (15.1%) and being told they had to pay for their vaccination by their general practitioner (5.8%).

## 40-Item knowledge questionnaire about influenza and vaccination

A total of 145 nursing students were purposively selected to participate in a pre- and post-knowledge questionnaire about influenza after receiving access to the 'serious game'. This sample was selected to determine the extent to which the game improved knowledge about influenza in a homogenous group. From these 124 nursing students (85.52%) completed a 40-item pre and post questionnaire. The questionnaire comprised of 27 true or false questions and 13 multiple choice questions, each with four available answers. Participants scored 1 point for every question correctly answered.

Overall, nursing students scored an average of 68.6% before receiving access to the game and 85.2% after, demonstrating a statistically significant increase (p<0.001) using paired t-tests. The Pearson's correlation coefficient was 0.594, indicating a moderate positive relationship between nursing student's knowledge about influenza and playing the game.

The most significant increases in knowledge that were noted post-game related to the amount of time it took to become fully protected from the influenza after the vaccination (10–14 days); the coverage of the influenza vaccination in relation to Australian Flu and Swine Flu; that most people who have influenza do not have symptoms, and that vaccination is recommended for pregnant women. The increase in knowledge across these questions ranged from 47.2%-70.3% post-test.

Increases in knowledge was recorded across 34 of the 40 items. The remaining six items demonstrated a marginal decrease of 3.6%-1.4%

Descriptive statistics from this 40-item questionnaire can be viewed in Table 2 and the dataset can be viewed in the supporting information.

## Discussion

To our knowledge, this is the first study that has examined the impact of a 'serious game' on nursing student knowledge and vaccination uptake. Pre-intervention, 36.7% of the sample had received the influenza vaccination in the preceding season. This low vaccination uptake is reflective of other research studies which have examined influenza vaccination uptake amongst this population group [31–35]. Post-intervention, vaccination uptake increased to 47.8% amongst the sample. While this modest increase is encouraging, it is evident that despite improvement in the attitudes and knowledge about influenza, more than half of the sample did not go on to receive their vaccination. Due to the nature of this study, it is difficult to judge the extent to which the serious game led to an increase in uptake. The study invitation, email reminders and clinical experiences of nursing students during this research are acknowledged as possible confounders.

The main reasons why nursing students did not receive their influenza vaccine were lack of time, uncertainty of where to receive the vaccination and concerns about receiving the vaccination. These reasons are reflective of the international literature on the reasons why qualified healthcare professionals do not receive influenza vaccination [17–26]. While there are no recommended vaccination uptake targets for nursing students, the Public Health Agency in England recently set the target of having 90% of its healthcare professional workforce vaccinated [51]. The target is 75% in Europe and the USA [5,6]. These figures would suggest that influenza vaccination of nursing students is a priority area due to their absence in influenza vaccination guidelines, empirical research and literature, combined with their apparent low uptake of the vaccination.

There has been a plethora of research studies which have demonstrated that provision of education, to address professional misconceptions about the influenza vaccination, have been associated with improvement in knowledge and subsequent uptake [26–30]. This study was reflective of this literature and provided all nursing students with six-months access to a 'serious game' about influenza. The students who participated in this study perceived their knowledge, likelihood to get the vaccination and likelihood to recommend the vaccination to patients and the public had improved after playing the game. In terms of the 40-item knowledge questionnaire, provided to a cohort of year one nursing students, we found highly statistically significant changes in level of knowledge after receiving access to the 'serious game'. Incidentally, this cohort of nursing students did not receive any additional education about influenza or influenza vaccination during their nursing programme in the six-month period

**Table 2. 40-Item knowledge questionnaire about influenza and vaccination.**

| Questionnaire Item | Pre-Test Score (% Correct Answers) | Post-Test Score (% Correct Answers) | Difference (% Correct Answers) |
|---|---|---|---|
| Influenza isn't such a big deal | 92.8% | 93.8% | 1.0% |
| The flu vaccine gives you flu | 78.4% | 93.8% | 15.4% |
| Which of these treatments does not treat influenza? (antibiotics, antiviral medications, influenza vaccination, keeping hydrated & staying warm) | 70.5% | 86.0% | 15.5% |
| Healthy people don't get seasonal flu | 95.0% | 93.0% | -2.0% |
| Flu is a mild illness so I don't need a vaccine | 93.5% | 93.8% | 0.3% |
| The side-effects of the flu vaccine are bad | 82.0% | 80.6% | -1.4% |
| How often should you get the flu jab? | 92.1% | 96.1% | 4.0% |
| The flu jab is safe | 95.7% | 93.8% | -1.9% |
| The influenza vaccine has been properly tested | 88.5% | 90.7% | 2.2% |
| How effective can the flu vaccine be? | 82.7% | 89.9% | 7.2% |
| Can pregnant women receive the vaccine? | 38.8% | 86.0% | 47.2% |
| You must avoid other people after receiving the vaccine because you'll be infectious | 93.5% | 89.9% | -3.6% |
| Should I go to work if I come into contact with someone who has flu? | 17.3% | 65.1% | 47.8% |
| Everyone should get the flu vaccine? | 36.7% | 38.0% | 1.3% |
| Where can you get the flu vaccine? | 69.1% | 85.3% | 16.2% |
| Once you've had the flu vaccine you are protected for life | 95.7% | 93.8% | -1.9% |
| This year's flu vaccine protects me against swine flu | 25.2% | 82.9% | 57.7% |
| Children can have the flu vaccine | 87.1% | 91.5% | 4.4% |
| When is the best time to get the flu vaccine? | 64.7% | 89.9% | 25.2% |
| It's nearly Christmas and I haven't had the flu vaccine. Is it now too late? | 85.6% | 91.5% | 5.9% |
| Vitamin C can prevent influenza | 62.6% | 85.3% | 22.7% |
| Breastfeeding mothers shouldn't have the influenza vaccine | 67.6% | 77.5% | 9.9% |
| I had the flu vaccine last year and do not need it again this year. | 91.4% | 94.6% | 3.2% |
| I am on antibiotics. Can I still have the flu jab? | 50.4% | 66.7% | 16.3% |
| How quickly are you protected after the flu vaccine? | 16.5% | 86.8% | 70.3% |
| I have a cold. Can I get the flu vaccine? | 48.2% | 89.1% | 40.9% |
| The flu vaccine makes it easier to catch other things like pneumonia. | 79.1% | 79.8% | 0.7% |
| Even healthy people can die from flu | 92.1% | 93.0% | 0.9% |
| Most people infected with influenza have no symptoms | 38.1% | 86.0% | 47.9% |
| I'm healthy and never catch flu, should I still get vaccinated? | 51.1% | 69.8% | 18.7% |
| Eating well and washing my hands will protect me from the flu | 44.6% | 77.5% | 32.9% |
| You can get the flu by going out in the cold without a coat | 64.0% | 87.6% | 23.6% |
| You must see your GP if you have flu | 54.7% | 79.8% | 25.1% |
| Fever, aches, exhaustion and a cough are all flu symptoms | 95.0% | 92.2% | -2.8% |
| Is my partner eligible for a free flu jab too? | 49.6% | 73.6% | 24.0% |
| People with underlying health conditions should get the flu vaccine? | 88.5% | 89.1% | 0.6% |
| The flu vaccine will protect me from the Australian Flu | 21.6% | 85.3% | 63.7% |
| My employer can force me to get the influenza vaccine | 82.0% | 92.2% | 10.2% |
| All nursing students should get the influenza vaccine | 84.2% | 91.5% | 7.3% |
| You can catch the flu from someone sneezing near you | 77.0% | 83.7% | 6.7% |
| **Average Overall Score** | **68.6%** | **85.2%** | **16.6%** |

they had access to the 'serious game'. The use of 'serious games' has already been associated with significant improvement in participant knowledge in previous studies [38–48]. Despite these positive findings, vaccine uptake remained below half after this study. While education

appears to be a supportive factor for increasing influenza vaccination uptake, providing an environment where nursing students can easily receive the influenza vaccine appears equally important. In the United Kingdom, front line healthcare professionals often receive their influenza vaccination at their place of work, for example at a staff influenza session within their hospital [15]. This was not the case for the nursing students included in this sample because many were attending university during the flu vaccination season, were not attending a clinical placement and therefore were responsible for organising their own vaccination with their general practitioner or local pharmacy.

While this study focused on influenza, the findings are also of interest in the context of the COVID-19 pandemic. Presently several COVID-19 vaccines are currently in human trials or have been rolled out. Despite the availability of COVID-19 vaccination, recent international research has suggested a high potential for vaccine hesitancy amongst the global population [52]. Vaccine hesitancy is now a global concern and presents a substantial obstacle to achieving community immunity from COVID-19 [53]. Governments, healthcare services and patient advocacy groups are now tasked with building vaccine literacy amongst all members of society and this research suggests that the use of gamification may be one such supportive evidence-based strategy.

There were some limitations to this study. While the study sample was large comparable to similar research, approximately two in three students did not participate. In addition, our sample was not equally spread, with more than half of our sample made up from year one nursing students. This makes generalisability of findings to similar settings difficult. With consideration to knowledge, this study only examined this in year one students, and it is therefore difficult to determine the impact of the serious game on knowledge of year 2 and year 3 nursing students. The study design would have also been strengthened had the serious game been compared to an alternative intervention or control. Finally, this study may have also been strengthened had it used validated instruments that have previously been used to measure attitudes to influenza and vaccination-related knowledge. Despite these limitations, this study makes a key contribution to a limited evidence-base and demonstrates how provision of a 'serious game' is very likely to improve knowledge of influenza and subsequent uptake of the vaccination.

## Conclusions

The research highlights the importance of equipping nursing students with education about influenza. Improvements in influenza knowledge is likely to encourage more nursing students to receive the influenza vaccination, an action that will help prevent influenza transmission between healthcare professionals and patients. This study also demonstrates how provision of a 'serious game' can provide nursing students with an innovative learning tool which has been associated with highly statistically significant improvements in knowledge.

## Supporting information

**S1 Dataset.**
(XLSX)

## Acknowledgments

We wish to thank the nursing students who participated in this study and Focus Games Ltd who agreed for us to carry out this independent evaluation and publish the findings prior to commencing data collection.

## Declarations

All authors meet at least one of the following criteria (recommended by the ICMJE: http://www.icmje.org/ethical_1author.html) and have agreed on the final version.

## Author Contributions

**Conceptualization:** Gary Mitchell, Laurence Leonard, Gillian Carter, Christine Brown Wilson.

**Formal analysis:** Gary Mitchell, Gillian Carter, Olinda Santin, Christine Brown Wilson.

**Investigation:** Gary Mitchell, Laurence Leonard, Gillian Carter, Christine Brown Wilson.

**Methodology:** Gary Mitchell, Laurence Leonard, Gillian Carter, Olinda Santin, Christine Brown Wilson.

**Project administration:** Gary Mitchell, Olinda Santin.

**Writing – original draft:** Gary Mitchell, Laurence Leonard, Christine Brown Wilson.

**Writing – review & editing:** Gary Mitchell, Laurence Leonard, Gillian Carter, Olinda Santin, Christine Brown Wilson.

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
