## [Decision Letter · Decision Letter 0]

11 Nov 2020

PONE-D-20-24096

Evaluation of a ‘serious game’ on nursing student knowledge and uptake of influenza vaccination.

PLOS ONE

Dear Dr. Mitchell,

Thank you for submitting your manuscript to PLOS ONE. After careful consideration, we feel that it has merit but does not fully meet PLOS ONE’s publication criteria as it currently stands. Therefore, we invite you to submit a revised version of the manuscript that addresses the points raised during the review process.

We look forward to receiving your revised manuscript.

Kind regards,

Mariusz Duplaga, Ph.D., M.D., Ass. Prof.

Academic Editor

PLOS ONE

Journal Requirements:

2.Thank you for including your ethics statement:

This study was approved by the School of Nursing and Midwifery’s Research Ethics Committee at Queen’s University Belfast in July 2018.

Please provide additional details regarding participant consent. In the ethics statement in the Methods and online submission information, please ensure that you have specified (1) whether consent was informed and (2) what type you obtained (for instance, written or verbal, and if verbal, how it was documented and witnessed). If your study included minors, state whether you obtained consent from parents or guardians. If the need for consent was waived by the ethics committee, please include this information.

Reviewers' comments:

Reviewer's Responses to Questions

**Comments to the Author**

1. Is the manuscript technically sound, and do the data support the conclusions?

Reviewer #1: Yes

Reviewer #2: Partly

2. Has the statistical analysis been performed appropriately and rigorously? 

Reviewer #1: Yes

Reviewer #2: Yes

3. Have the authors made all data underlying the findings in their manuscript fully available?

Reviewer #1: Yes

Reviewer #2: Yes

4. Is the manuscript presented in an intelligible fashion and written in standard English?

Reviewer #1: Yes

Reviewer #2: Yes

5. Review Comments to the Author

Reviewer #1: The study describes the evaluation with a pre- and post-design of a serious games aimed to increase knowledge about influenza vaccination (and, consequently, vaccination uptake) among nursing students. The paper is very well written and the study is timely.

Major comment:

- The design of the study is a bit complex. You had two homogenous cohorts, but they were not really comparable since they were not administered the same questionnaires. Is this the case? This might be a limitation and should be stated as such in the Discussion. If I misunderstood the design of the study and the completion of the questionnaires, this means that the description is not clear and that the Methods section should be better written. The reader gets lost with the different questionnaires, cohorts, timeline… Overall, it is not clear why you recruited two cohorts.

Minor comments:

- Since you have the chance to revise your paper before publication, I suggest to mention in either the Introduction or the Discussion Covid-19. This would add some practical implications from your study to the current pandemics’ situation.

- Please justify why you have created ad-hoc questionnaires to measure attitudes to influenza and vaccination-related knowledge even if some validated instruments already exist.

- In the Introduction you mention that in the UK vaccinations are administered to 70% front-line healthcare professionals. Do you have access to any data about real uptake? The percentage of 70% is quite high, especially compared to other European countries. Please confirm this high uptake.

- Include p values in Table 1.

- Lack of time is the main reason for not getting vaccinated. This could be better addressed in your Discussion. It seems that knowledge is not that important to get vaccinated…

- In Table 2, provide the answers for “Which of these treatments does not treat influenza”. This stand-alone item is difficult to interpret.

Reviewer #2: This is a potentially interesting study on the impact of a serious game intervention on influenza vaccination. The paper's main strength is the fact that it seems to be the first to assess this type of intervention in the context of influenza vaccination. It's main weaknesses are the potential confounders that make it hard to ascertain a causal link between exposure to the game and the observed increase in vaccination uptake, and the lack of comparison to alternative (possibly cheaper and less time-consuming than a serious game) interventions.

In general, the paper is well written and the study's findings are clear. The methods section is also clear, but the subsection on the intervention should have described the game in more detail. A better description of the game would help the reader assess possible confounding factors, and form a better idea of what other kinds of intervention might be comparable to the intervention assessed in this study.

The soundness of attitudes and perception questionnaire is debatable. I found it doubtful that the respondent could have formed an unbiased opinion of the state of their knowledge and attitude regarding vaccination after playing the game. The authors should discuss this point.

It also strikes me that it is hard to judge the extent to which any change of attitude, reflected in the higher vaccination uptake, is due to the game, or to other factors surrounding the study. It could be due, for instance, to the pre-study questionnaire acting as a reminder or raising awareness to vaccination, irrespectively of the game intervention. This issue should be discussed. I also found it somewhat surprising that the authors did not include an alternative intervention for comparison. Might a leaflet or a short video have been as effective as the serious game?

6. PLOS authors have the option to publish the peer review history of their article (what does this mean?). If published, this will include your full peer review and any attached files.

Reviewer #1: **Yes: **Ilaria Montagni

Reviewer #2: **Yes: **Saturnino Luz

---

## [Author Response · Author response to Decision Letter 0]

21 Dec 2020

This has been uploaded as a separate document with this submission. For convenience it is also noted below:

Response to Reviewers

PONE-D-20-24096: Evaluation of a ‘serious game’ on nursing student knowledge and uptake of influenza vaccination. PLOS ONE

Journal Requirements:

 Thank you for this. We have now reviewed these style requirements and amended our submission accordingly.

2.Thank you for including your ethics statement:

This study was approved by the School of Nursing and Midwifery’s Research Ethics Committee at Queen’s University Belfast in July 2018.

Please provide additional details regarding participant consent. In the ethics statement in the Methods and online submission information, please ensure that you have specified (1) whether consent was informed and (2) what type you obtained (for instance, written or verbal, and if verbal, how it was documented and witnessed). If your study included minors, state whether you obtained consent from parents or guardians. If the need for consent was waived by the ethics committee, please include this information.

Thank you. We have included the following additional information about consent: Participants did not provide verbal or written consent but were informed that they were under no obligation to complete any of the questionnaires. Participants gave their consent to complete the questionnaire when they actively accessed the survey web links. 

Review Comments to the Author

Reviewer #1: The study describes the evaluation with a pre- and post-design of a serious games aimed to increase knowledge about influenza vaccination (and, consequently, vaccination uptake) among nursing students. The paper is very well written and the study is timely.

Major comment:

- The design of the study is a bit complex. You had two homogenous cohorts, but they were not really comparable since they were not administered the same questionnaires. Is this the case? This might be a limitation and should be stated as such in the Discussion. If I misunderstood the design of the study and the completion of the questionnaires, this means that the description is not clear and that the Methods section should be better written. The reader gets lost with the different questionnaires, cohorts, timeline… Overall, it is not clear why you recruited two cohorts.

Thank you for this feedback. We have clarified this in the manuscript and provided clearer rationale about why a specific cohort of participants (year one students) was selected and noted this as a limitation in the manuscript as they are not comparable with the first cohort of participants (that included year 1, 2 and 3 students).

Minor comments:

- Since you have the chance to revise your paper before publication, I suggest to mention in either thm e Introduction or the Discussion Covid-19. This would add some practical implications from your study to the current pandemics’ situation.

Thank you. We have included information about COVID-19 and the potential of translating this approach within the discussion section as suggested: While this study focused on influenza, the findings are also of interest in the context of the COVID-19 pandemic. Presently several COVID-19 vaccines are currently in human trials or have been rolled out. Despite the availability of COVID-19 vaccination, recent international research has suggested a high potential for vaccine hesitancy amongst the global population[52]. Vaccine hesitancy is now a global concern and presents a substantial obstacle to achieving community immunity from COVID-19[53]. Governments, healthcare services and patient advocacy groups are now tasked with building vaccine literacy amongst all members of society and this research suggests that the use of gamification may be one such supportive evidence-based strategy.

- Please justify why you have created ad-hoc questionnaires to measure attitudes to influenza and vaccination-related knowledge even if some validated instruments already exist. 

Thank you for this comment. We have included the following paragraph to provide clarity: While other validated measures were available, the authors developed their own questionnaires to answer the research question. The rationale for this was due to a combination of factors including the nature of the intervention (a digital game), its duration (each play taking approximately two minutes), the sample (nursing students) and the information the authors wanted to glean.

We have noted the following within the limitations section of the manuscript: Finally, this study may have also been strengthened had it used validated instruments that have previously been used to measure attitudes to influenza and vaccination-related knowledge. 

- In the Introduction you mention that in the UK vaccinations are administered to 70% front-line healthcare professionals. Do you have access to any data about real uptake? The percentage of 70% is quite high, especially compared to other European countries. Please confirm this high uptake. 

We have rechecked this reference and this was correct (for England – not the whole UK: (739,187/1,051,851 of front line healthcare workers: 70.3%). We have taken the opportunity to update our statistics for England and this is now 74.3% (791,112/1,065,017).

- Lack of time is the main reason for not getting vaccinated. This could be better addressed in your Discussion. It seems that knowledge is not that important to get vaccinated…

Thank you. We have included the following: Despite these positive findings, vaccine uptake remained below half after this study. While education appears to be a supportive factor for increasing influenza vaccination uptake, providing an environment where nursing students can easily receive the influenza vaccine appears equally important. In the United Kingdom, front line healthcare professionals often receive their influenza vaccination at their place of work, for example at a staff influenza session within their hospital[15]. This was not the case for the nursing students included in this sample because many were attending university during the flu vaccination season, were not on a clinical placement and therefore were responsible for organising their own vaccination with their general practitioner or local pharmacy.

- In Table 2, provide the answers for “Which of these treatments does not treat influenza”. This stand-alone item is difficult to interpret.

Thank you for this comment. We have included the options to this question: Antibiotics, antiviral medications, influenza vaccination or keeping hydrated/staying warm.

Reviewer #2: This is a potentially interesting study on the impact of a serious game intervention on influenza vaccination. The paper's main strength is the fact that it seems to be the first to assess this type of intervention in the context of influenza vaccination. It's main weaknesses are the potential confounders that make it hard to ascertain a causal link between exposure to the game and the observed increase in vaccination uptake, and the lack of comparison to alternative (possibly cheaper and less time-consuming than a serious game) interventions.

Thank you for your review. We have included the following sentence in our limitations section in response to your feedback: The study design would have also been strengthened had the serious game been compared to an alternative intervention or control.

In general, the paper is well written and the study's findings are clear. The methods section is also clear, but the subsection on the intervention should have described the game in more detail. A better description of the game would help the reader assess possible confounding factors, and form a better idea of what other kinds of intervention might be comparable to the intervention assessed in this study.

Thank you for this comment. We have now provided the following information in this section: . Players of the Flu Bee Game can share their success on the game’s leader board and invite colleagues to play via social media. The serious game presents players with influenza facts and challenges common myths associated with vaccine hesitancy. These common myths include statements like “I’m healthy and so do not need a flu vaccine”, “I’m 12 weeks pregnant so cannot have the flu vaccine” or “I don’t want to risk getting the flu from the vaccine” [49-50]. The overall objective of this serious game is to create awareness about influenza, dispel myths associated with the influenza vaccine and increase uptake the vaccination.

The soundness of attitudes and perception questionnaire is debatable. I found it doubtful that the respondent could have formed an unbiased opinion of the state of their knowledge and attitude regarding vaccination after playing the game. The authors should discuss this point.

Thank you for this comment. We have included the following text in our limitations section with regards to the attitudes/perception questionnaire: Finally, this study may have also been strengthened had it used validated instruments that have previously been used to measure attitudes to influenza and vaccination-related knowledge

It also strikes me that it is hard to judge the extent to which any change of attitude, reflected in the higher vaccination uptake, is due to the game, or to other factors surrounding the study. It could be due, for instance, to the pre-study questionnaire acting as a reminder or raising awareness to vaccination, irrespectively of the game intervention. This issue should be discussed. I also found it somewhat surprising that the authors did not include an alternative intervention for comparison. Might a leaflet or a short video have been as effective as the serious game?

Thank you for this helpful comment. We have now acknowledged the potential confounders in the discussion: Due to the nature of this study, it is also difficult to judge the extent to which the serious game led to an increase in uptake. The study invitation, email reminders and clinical experiences of nursing students during this research are acknowledged as possible confounders. 

In terms of the control/comparator we have included the following in our limitations section: The study design would have also been strengthened had the serious game been compared to an alternative intervention or control

Thank you to the reviewers for taking the time to review this manuscript and providing their helpful feedback. We believe this process has supported us in strengthening the manuscript.

---

## [Editor Report · Decision Letter 1]

30 Dec 2020

Evaluation of a ‘serious game’ on nursing student knowledge and uptake of influenza vaccination.

PONE-D-20-24096R1

Dear Dr. Mitchell,

We’re pleased to inform you that your manuscript has been judged scientifically suitable for publication and will be formally accepted for publication once it meets all outstanding technical requirements.

Kind regards,

Mariusz Duplaga, Ph.D., M.D., Ass. Prof.

Academic Editor

PLOS ONE

---

## [Editor Report · Acceptance letter]

6 Jan 2021

PONE-D-20-24096R1 

Evaluation of a ‘serious game’ on nursing student knowledge and uptake of influenza vaccination. 

Dear Dr. Mitchell:

I'm pleased to inform you that your manuscript has been deemed suitable for publication in PLOS ONE. Congratulations! Your manuscript is now with our production department. 

Kind regards, 

on behalf of

Dr. Mariusz Duplaga 

Academic Editor

PLOS ONE